# AttCAT: Explaining Transformers via Attentive Class Activation Tokens

**Yao Qiang    Deng Pan    Chengyin Li    Xin Li    Rhongho Jang    Dongxiao Zhu**
Department of Computer Science, Wayne State University
{yao,pan.deng,cyli,xinlee,r.jang,dzhu}@wayne.edu

## Abstract

Transformers have improved the state-of-the-art in various natural language processing and computer vision tasks. However, the success of the Transformer model has not yet been duly explained. Current explanation techniques, which dissect either the self-attention mechanism or gradient-based attribution, do not necessarily provide a faithful explanation of the inner workings of Transformers due to the following reasons: first, attention weights alone without considering the magnitudes of feature values are not adequate to reveal the self-attention mechanism; second, whereas most Transformer explanation techniques utilize self-attention module, the skip-connection module, contributing a significant portion of information flows in Transformers, has not yet been sufficiently exploited in explanation; third, the gradient-based attribution of individual feature does not incorporate interaction among features in explaining the model's output. In order to tackle the above problems, we propose a novel Transformer explanation technique via attentive class activation tokens, aka, AttCAT, leveraging encoded features, their gradients, and their attention weights to generate a faithful and confident explanation for Transformer's output. Extensive experiments are conducted to demonstrate the superior performance of AttCAT, which generalizes well to different Transformer architectures, evaluation metrics, datasets, and tasks, to the baseline methods. Our code is available at: https://github.com/qiangyao1988/AttCAT.

## 1 Introduction

Transformers have advanced the state-of-the-art on a variety of natural language processing tasks [1, 2] and see increasing popularity in the field of computer vision [3, 4]. The main innovation behind the Transformer models is the stacking of multi-head self-attention layers to extract global features from sequential tokenized inputs. However, the lack of understanding of their mechanism increases the risk of deploying them in real-world applications [5, 6, 7, 8, 9]. This has motivated new research on explaining Transformers output to assist trustworthy human decision-making [10, 11, 12, 13, 14, 15, 16, 17].

The self-attention mechanism [18] in Transformers assigns a pairwise score capturing the relative importance between every two tokens or image patches as attention weights. Thus, a common practice is to use these attention weights to explain the Transformer model's output by exhibiting the importance distribution over the input tokens [6]. The baseline method, shown as RawAtt in Figure 2, utilizes the raw attention weights from a single layer or a combination of multiple layers [10]. However, recent studies [11, 12, 13] question whether highly attentive inputs significantly impact the model outputs. Serrano et al. [11] demonstrate that erasing the representations accorded high attention weights do not necessarily lead to a performance decrease. Jain et al. [12] suggest that "attention is not explanation" by observing that attention scores are frequently inconsistent with other feature importance indicators like gradient-based measures. Abnar et al. [13] argue that the contextual information from tokens gets more similar as going deeper into the model, leading to unreliable

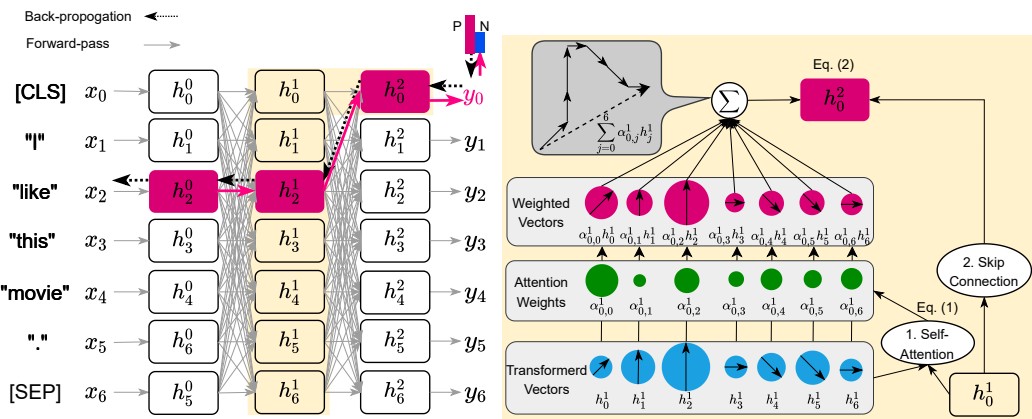

Figure 1: An illustration of Transformer architecture. The left panel shows a simple three-layer Transformer model. Each layer consists of a self-attention module and a skip connection module (shown in the right panel). The input is a sequence of tokens with two added special tokens, i.e., [CLS] and [SEP]. The third token, 'like' ($x_2$), contributes mostly to the positive sentiment prediction since its attention weighted output is the largest. Size of the colored circles illustrate the value of the scalar or the norm of the corresponding vector. Arrows within the circles demonstrate the directions of the vectors.

explanations using the raw attention weights. The authors propose two methods to combine the attention weights across multiple layers to cope with this issue. Their attention rollout method, shown as Rollout in Figure 2, reassigns the important scores to the tokens through the linear combination of attention weights across the layers tracing the information flow in Transformer. However, the rollout operation canceled out the accumulated important scores as some deeper layers have almost uniformly distributed attention weights. The attention flow method is formulated as a max-flow problem by dissecting the graph of pairwise attentions. While it somewhat outperforms the rollout method in specific scenarios, it is not ready to support large-scale evaluations [15].

Recently, Bastings et al. [19] advocate using saliency method as opposed to attention as explanations. Although some gradient-based methods [20, 21, 22, 23] have been proposed to leverage salience for explaining Transformer's output, most of them still focus on the gradients of attention weights, i.e., Grads and AttGrads as shown in Figure 2. They suffer from a similar limitation to the above-mentioned attention-based methods. Layer-wise Relevance Propagation (LRP) method [24, 25], which is also considered as a type of saliency method, propagates relevance scores from the output layer to the input. There has been a growing body of work on using LRP to explain Transformers [14, 15]. Voita et al. [14] use LRP to capture the relative importance of the attention heads within each Transformer layer (shown as PartialLRP in Figure 2). However, this approach is limited by only providing partial information on each self-attention head's relevance; no relevance score is propagated back to the input. To address this problem, Chefer et al. [15] provide a comprehensive treatment of the information propagation within all components of the Transformer model, which back-propagates the information through all layers from the output back to the input. This method further integrates gradients from the attention weights, shown as TransAtt in Figure 2. However, TransAtt relies on the specific LRP rules that is not applicable for other attention modules, e.g., co-attention. Thus it can not provide explanations for all transformer architectures [26].

As such, the existing Transformer explanation techniques are not completely satisfactory due to three major issues. First, most attention-based methods disregard the magnitudes of the features. The summation operation (Eq. 2 shown in Figure 1) demonstrates both attention weights (the green circles) and the feature (the blue circles) contribute to the weighted outputs (the red circles). In other words, since the self-attention mechanism involves the computation of queries, keys, and values, reducing it only to the derived attention weights (inner products of queries and keys) is not ideal. Second, besides the self-attention mechanism, skip connection as another major component in Transformer is not even considered in current techniques. The latter enables the delivery and integration of information by adding an identity mapping from inputs to outputs, trying to solve the model optimization problem from the perspective of information transfer [27]. Moreover, Lu et al. [28] find that a significant

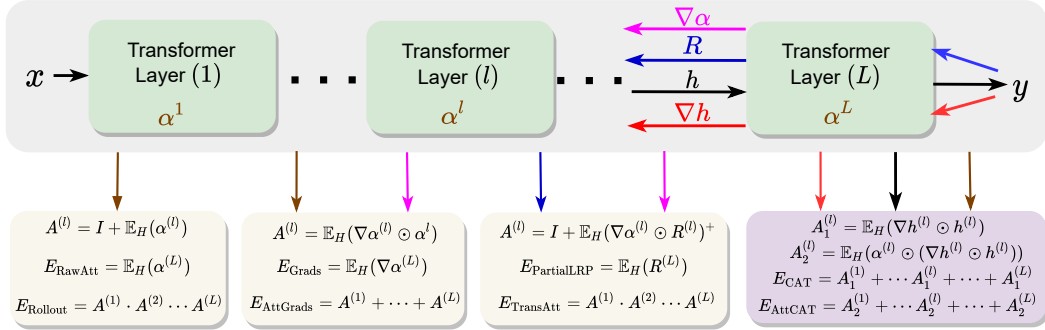

Figure 2: A summary of the existing explanation methods and our methods: CAT and AttCAT. The Transformer consists of several layers denoted as Layer $(1), \cdots, (l), \cdots, (L)$. $\nabla\alpha$ and $\nabla h$ represent the gradients of attention weights $\alpha$ and outputs $h$, respectively. $R$ is calculated based on layer-wise relevance propagation (LRP). $E$ denotes the explanation method. $\mathbb{E}_H$ means averaging among multi-head attentions in each layer.

portion of information flow in BERT goes through the skip connection instead of the attention heads (i.e., three times more often than attention on average). Thus, attention alone, without considering the skip connection, is not sufficient to characterize the inner working mechanism of Transformers. Third, the individual feature attribution-based approaches [15, 14, 29, 30] cannot capture the pairwise interactions of feature since gradients or relevance scores are calculated independently for each individual feature. For example, the gradients directly go through the Transformer layers from the output to the specific input (the token 'like'), shown in Figure 1.

We propose Attentive Class Activation Tokens (AttCAT) to generate token-level explanations leveraging features, their gradients, and their self-attention weights. Inspired by GradCAM [31], which uses gradient information flowing into the last convolutional layer of the Convolutional Neural Network (CNN) to understand the importance of each neuron for the decision of interest, our approach quantifies the impact of each token to the class-specific output via its gradient information. We further leverage the self-attention weights to capture the global contextual information of each token since it determines the relative importance of a single token concerning all other tokens in the input sequence. By disentangling the information flow across the Transformer layers for a specific token into the information from itself via a skip connection and the interaction information among all the tokens via a self-attention mechanism, we integrate the impact scores, which are generated using AttCAT, from multiple layers to give the final explanation.

A summary of the baseline methods and our AttCAT method is shown in Figure 2, demonstrating their main similarities and differences. The RawAtt and Rollout [13] methods simply use the attention weights ($\alpha$). The Grads method leverages the gradients of attention weights ($\nabla\alpha^L$) from the last Transformer layer, while the AttGrads method [22] integrates the attention weights ($\alpha$) and their gradients ($\nabla\alpha$) from all Transformer layers. The PartialLRP method [14] applies LRP only on the last Transformer layer ($R^L$). Differently, the TransAtt method [26] integrates the relevance scores ($R$) from LRP and the gradients of attention weights ($\nabla\alpha$). We use CAT, a new gradient-based attribution method leveraging the features ($h$) and their gradients ($\nabla h$), as our in-house baseline method. We further integrate attention weights ($\alpha$) with CAT as the proposed AttCAT method.

We state our contributions as follows:

- We propose a novel Transformer explanation technique, AttCAT, leveraging the features, their gradients together with attention weights to generate the so-called impact scores to quantify the influence of inputs on the model's outputs.

- Our AttCAT exploits both the self-attention mechanism and skip connection to explain the inner working mechanism of Transformers via disentangling information flows between intermediate layers.

- Furthermore, our class activation based method is capable of discriminating positive and negative impacts toward the model's output using the directional information of the gradients.
- Finally, we conduct extensive experiments on different Transformer architectures, datasets, and Natural Language Processing (NLP) tasks, demonstrating a more faithful and confident explanation than the baseline methods using several quantitative metrics and qualitative visualizations.

## 2 Preliminaries

### 2.1 Self-Attention Mechanism

The encoders in Transformer model [1] typically stack $L$ identical layers. Each contains two sub-layers: (a) a multi-head self-attention module and (b) a feed-forward network module, coupled with layer normalization and skip connection. As illustrated in Figure 1, each encoder computes the output $\mathbf{h}_i^{(l)} \in \mathbb{R}^d$ of the $i$-th token combining the previous encoder's corresponding output $\mathbf{h}_i^{(l-1)}$ from the skip connection and a sequence output $\mathbf{h}^{(l-1)} = \{\mathbf{h}_1^{(l-1)}, \cdots, \mathbf{h}_i^{(l-1)}, \cdots, \mathbf{h}_n^{(l-1)}\} \subseteq \mathbb{R}^d$ through self-attention mechanism:

$$\alpha_{i,j}^l := \mathrm{softmax}\left(\frac{Q(\mathbf{h}_i^{(l-1)})K(\mathbf{h}_j^{(l-1)})^T}{\sqrt{d}}\right) \in \mathbb{R}, \tag{1}$$

$$\mathbf{h}_i^l = \mathbf{W}^O\left(\sum_{j=1}^n \alpha_{i,j} V(\mathbf{h_j}^{(l-1)}) + \mathbf{h}_i^{(l-1)}\right), \tag{2}$$

where $\alpha_{i,j}^l$ is the attention weight assigned to the $j$-th token for computing $\mathbf{h}_i^{(l)}$. $d$ denotes the dimension of the vectors. Here, $Q(\cdot)$, $K(\cdot)$, and $V(\cdot)$ are the query, key, and value transformations:

$$Q(\mathbf{h}) := \mathbf{W}^Q \mathbf{h}, \;\; K(\mathbf{h}) := \mathbf{W}^K \mathbf{h}, \;\; V(\mathbf{h}) := \mathbf{W}^V \mathbf{h}, \;\; (\mathbf{W}^Q, \mathbf{W}^K, \mathbf{W}^V) \in \mathbb{R}^{d\times d}, \tag{3}$$

respectively. We drop the bias parameters in these equations for simplicity. For multi-head attentions, we concatenate the output from each head.

### 2.2 Class Activation Map

GradCAM [31] is one the most successful CAM-based methods using the gradient information flowing into the last convolutional layer of CNN to understand the importance of each neuron for the decision of interest. In order to obtain the class discriminative localization map for the explanation, Grad-CAM first computes the gradient of the score for class $c$, i.e., $y^c$ before the softmax, concerning feature maps $A^k$ of a convolutional layer as $\frac{\partial y^c}{\partial A^k}$. Then, these flowing back gradients are global-average-pooled to obtain the neuron importance weight $w_k^c$:

$$w_k^c = \mathbb{E}\left(\frac{\partial y^c}{\partial A^k}\right), \tag{4}$$

where $\mathbb{E}$ denotes the global-average-pooling. The weight $w_k^c$ reflects a partial linearization of the CNN downstream from $A$ and captures the importance of feature map $k$ for a target class $c$. Then a weighted combination of forward activation maps is obtained by:

$$\mathrm{GradCAM}^c = \mathrm{ReLU}\left(\sum_k w_k^c A^k\right), \tag{5}$$

where ReLU() is applied to filter out the negative values since we are only interested in the features that positively influence the class of interest.

## 3 Problem Formulation

The objective of a token-level explanation method for Transformer is to generate a separate score for each input token in order to answer the question: *Given an input text and a trained Transformer*

*model, which tokens mostly influence the model's output?* There is no standard definition of influence in literature [32]. Some works use the term 'importance', whereas others use the term 'relevance' depending on the explanation methods being used. Here we note that the token influence should reflect not only the magnitude of impact but also its directionality. As such, we define a new concept, Impact Score, to measure both **Magnitude of Impact** and **Directionality**. The former addresses the question "Which input tokens contribute mostly to the output?". And the latter addresses the question "Given an input token, have positive or negative contributions been made to the output?" Formally, we define the Impact Score generated by our AttCAT method as follows:

**Definition 1 (Impact Score)**   *Given a pre-trained Transformer $T(\cdot)$, an input token $x$, and our explanation method $E_{\text{AttCAT}}(\cdot)$. Impact Score is define as:*

$$\text{Impact Score}(E_{\text{AttCAT}}(T(x))) = \begin{cases} |E_{\text{AttCAT}}(T(x))|, & \text{Magnitude of Impact,} \\ \text{Sign}(E_{\text{AttCAT}}(T(x))), & \text{Directionality.} \end{cases} \tag{6}$$

**Remark 1 (Magnitude of Impact)**   The magnitude of impact indicates how much contribution has been made by each token. A sort function can be applied to the array of scores for the input tokens to retrieve the most impactful tokens on the output.

**Remark 2 (Directionality)**   The sign reveals whether each token makes a positive or negative impact on the output.

## 4   Attentive Class Activation Tokens

### 4.1   Disentangling Information Flows in Transformer

To interpret the inner working mechanism of Transformers, it is essential to understand how the information of each input token flows through each intermediate layer and finally reaches the output. Some previous works [13, 22] use heuristics to treat high attention weights and/or their gradients as indicators of important information flows across layers. Others [15, 14] apply LRP aiming to dissect the information flows via layer-wise back-propagation. However, these approaches either rely on the simple-but-unreliable assumption of linear combination of the intermediate layers or ignore the major components of Transformer, i.e., the magnitudes of the features and the skip connection.

From Figure 1, we observe that the output sequence of the Transformer model has a one-to-one correspondence to its input sequence. The skip connection is a shortcut that bridges the input and output of the self-attention operation. We note that the Transformer encoder intuitively is an operator that adds the representation of token interactions (via self-attention mechanism) onto the original representation of the token (via skip connection). Therefore, from a perspective of information flow, we can specify the $i$-th token's information at the ($l$)-th layer as:

$$\text{Information}(\mathbf{x}_i^l) = \text{Information}(\mathbf{x}_i^{l-1}) + \text{Interaction}(\mathbf{x}_i^{l-1}, \mathbf{x}_{n/i}^{l-1}), \tag{7}$$

where $\text{Information}(\mathbf{x}_i^{l-1})$ represents the information contained in the $i$-th token at the ($l$-1)-th layer, and $\text{Interaction}(\mathbf{x}_i^{l-1}, \mathbf{x}_{n/i}^{l-1})$ reflects the summation of all pairwise interaction between the $i$-th token and all other tokens ($n/i$).

This observation motivates us to interpret the inner working mechanism of Transformers via disentangling the information flow Transformer. Thus, considering Eq. 7 as a recurrence relation, the final representation of the $i$-th token then consists of the original information (the input) plus token interactions between the $i$-th token and all other tokens at different layers. Since the CNN's last convolutional layer also encodes both high-level semantics and detailed spatial information, corresponding to the original information and the interactions herein, the way GradCAM used for explaining a CNN model's output inspired us to design Attentive Class Activation Tokens (AttCAT) to understand the impact of each token on a Transformer model's output.

### 4.2   Class Activation Tokens

For a pre-trained Transformer, we can always find its output $\mathbf{h}^l$ at $l$-th layer. Assume $\mathbf{h}^l$ has $n$ columns, each column corresponds to an input token (including the paddings, i.e., [CLS] and [SEP]).

We write its columns separately as $\mathbf{h}_1^l, \cdots, \mathbf{h}_i^l, \cdots, \mathbf{h}_n^l$. As $\mathbf{h}_i^L$ is the output of $i$-th token from the last Transformer layer $L$, to interpret the impact of $i$-th token to the final output $y^c$ for class $c$, it would be straightforward if we have a linear relationship between $y^c$ and $\mathbf{h}_i^L$ as follows:

$$y^c = \sum_i^n \mathbf{w}_i^c \cdot \mathbf{h}_i^L, \tag{8}$$

where $\mathbf{w}_i^c$ is the linear coefficient vector for $\mathbf{h}_i^L$. Inspired by GradCAM [31], we obtain the token important weights as:

$$\mathbf{w}_i^c = \nabla \mathbf{h}_i^L = \frac{\partial y^c}{\partial \mathbf{h}_i^L}, \tag{9}$$

where $\mathbf{w}_i^c$ illustrates a partial linearization from $\mathbf{h}_i^L$ and captures the importance of $i$-th token to a target class $c$. Class Activation Tokens (CAT) is then obtained through a weighted combination:

$$\mathrm{CAT}_i^L = \nabla \mathbf{h}_i^L \odot \mathbf{h}_i^L, \tag{10}$$

where $\odot$ is the Hadamard product. $\mathrm{CAT}_i^L$ denotes the impact score of the $i$-th token at $L$-th layer towards class $c$. Note that we do not apply ReLU() to filter out the negative scores here since we also care about the directionality of the impact score.

### 4.3 Attentive CAT

While CAT explains the model's output according to the attribution of each individual token's encoder output (Eq. 8), it does not consider the interaction among tokens, which is revealed via the self-attention mechanism. The self-attention mechanism [18] assigns a pairwise similarity score between every two tokens as the attention weight, encoding the important interaction information of these tokens. Therefore, we integrate self-attention weights with CAT to further incorporate the token interaction information for better quantifying the impact of each token on the Transformer model's output. Our Attentive CAT (AttCAT) at $L$-th layer for $i$-th token is then formulated as:

$$\mathrm{AttCAT}_i^L = \mathbb{E}_H(\alpha_i^L \cdot \mathrm{CAT}_i^L), \tag{11}$$

where $\alpha_i^L$ denotes the attention weights of the $i$-th token at $L$-th layer. $\mathbb{E}_H(\cdot)$ means averaging over multiple heads.

Recall that Eq. 7 represents a recurrence relation, we can always find the output of $l$-th layer and assign it as $y_i^l$. We can use Eq. 9, 10, and 11 to formulate $\mathrm{AttCAT}_i^l$, denoting the impact score for $i$-th token at $l$-th layer.

Finally, different from the Rollout and TransAtt methods that apply the rollout operation, we sum $\mathrm{AttCAT}_i^l$ over all Transformer layers as the final impact score of $i$-th token as follows:

$$\mathrm{AttCAT}_i = \sum_{j=1}^L \mathrm{AttCAT}_i^j. \tag{12}$$

We empirically demonstrate that the summation is a more effective way than Rollout in Figure 5.

## 5 Experiments

### 5.1 Desirable Properties of an Explanation Technique

We first introduce two desirable properties of an explanation method: faithfulness and confidence, along with metrics to systematically evaluate the performance of various explanation techniques.

**Faithfulness** quantifies the fidelity of an explanation technique by measuring if the tokens identified indeed impact the output. We adopt two metrics from prior work to evaluate the faithfulness of word-level explanations: the area over the perturbation curve (AOPC) [33, 34] and the Log-odds scores [35, 34]. These two metrics measure local fidelity by deleting or masking the top $k\%$ scored words and comparing the probability change on the predicted label.

**Confidence** A token can receive several saliency scores, indicating its contribution to the prediction of each class. The tokens with higher impact scores of the predicted class $c$ should also have lower

impact scores for the remaining classes. In other words, the explanation techniques should be highly confident in recognizing the most impact tokens of the desired class (usually the predicted class). On the other hand, these tokens should have the most negligible impact on other classes. We use Kendall-$\tau$ correlation, the statistic measuring the strength of association between the ranked scores of different classes, to evaluate the confidence of an explanation method.

## 5.2 Experiment Settings

**Transformer models:** BERT [2] is one of the most representative Transformer models with impressive performance across a variety of NLP tasks, e.g., sentiment analysis and question answering. We use the $\text{BERT}_{\text{base}}$ model and some variants (i.e., DistillBERT [36] and RoBERTa [37]) in our experiments. Our method can be generally applied to other Transformer architectures with minor modifications. The pre-trained models from Huggingface[1] are used for validating our explanation method and comparing it to others. More details of these Transformer models and their prediction performance are presented in Appendix A.

**Datasets:** We evaluate the performance using the following exemplar tasks: sentiment analysis on SST2 [38] , Amazon Polarity, Yelp Polarity [39], and IMDB [40] data sets; natural language inference on MNLI [41] data set; paraphrase detection on QQP [42] data set; and question answering on SQuADv1 [43] and SQuADv2 [44] data sets. More details of these data sets are described in Appendix B.

**Baseline methods:** Several baseline explanation methods for Transformer have been compared through our experiments, including the attention-based methods (i.e., RawAtt and Rollout [13]), the attention gradient-based methods (i.e., Grads and AttGrads [22]), the LRP-based methods (i.e., PartialLRP [14] and TransAtt [15]). CAT without incorporating attention weights is an ablation version of AttCAT. Figure 2 summarizes and compares these methods with formulations.

## 5.3 Evaluation Metrics

**AOPC:** By deleting top $k\%$ words, AOPC calculates the average change of the prediction probability on the predicted class over all test examples as follows:

$$\text{AOPC}(k) = \frac{1}{N} \sum_{i=1}^{N} p(\hat{y}|\mathbf{x}_i) - p(\hat{y}|\tilde{\mathbf{x}}_i^k), \tag{13}$$

where $N$ is the number of examples, $\hat{y}$ is the predicted label, $p(\hat{y}|\cdot)$ is the probability on the predicted class, and $\tilde{\mathbf{x}}_i^k$ is constructed by removing the $k\%$ top-scored words from $\mathbf{x}_i$. To avoid choosing an arbitrary $k$, we remove $0, 10, 20, \cdots, 100\%$ of the tokens in order of decreasing saliency, thus arriving at $\tilde{\mathbf{x}}_i^0, \tilde{\mathbf{x}}_i^{10}, \cdots, \tilde{\mathbf{x}}_i^{100}$. Higher values of AOPC are better, which means the deleted words are more impactful on the model's output.

**LOdds:** Log-odds score is calculated by averaging the difference of negative logarithmic probabilities on the predicted class over all test examples before and after masking $k\%$ top-scored words with zero paddings,

$$\text{LOdds}(k) = \frac{1}{N} \sum_{i=1}^{N} \log \frac{p(\hat{y}|\tilde{\mathbf{x}}_i^k)}{p(\hat{y}|\mathbf{x}_i)}. \tag{14}$$

The notations are the same as in Eq. 13 with the only difference that $\tilde{\mathbf{x}}_i^k$ is constructed by replacing the top $k\%$ word with the special token [PAD] in $\mathbf{x}_i$. Lower LOdds scores are better.

**Kendal correlation:** We use the Kendal-$\tau$ to evaluate confidence of an explanation method, formally:

$$\text{Kendal correlation} = \frac{1}{N} \sum_{i=1}^{N} \text{Kendall-}\tau(S(\mathbf{x}_i)_c, S(\mathbf{x}_i)_{C/c}), \tag{15}$$

where $S(\mathbf{x}_i)$ denotes an array of the token index in order of the decreasing saliency (or attribution, or relevance, or impact) scores for a test example. A lower Kendal correlation demonstrates the explanation method is more confident in generating the saliency scores for predicting the class $c$.

---

[1] https://huggingface.co/

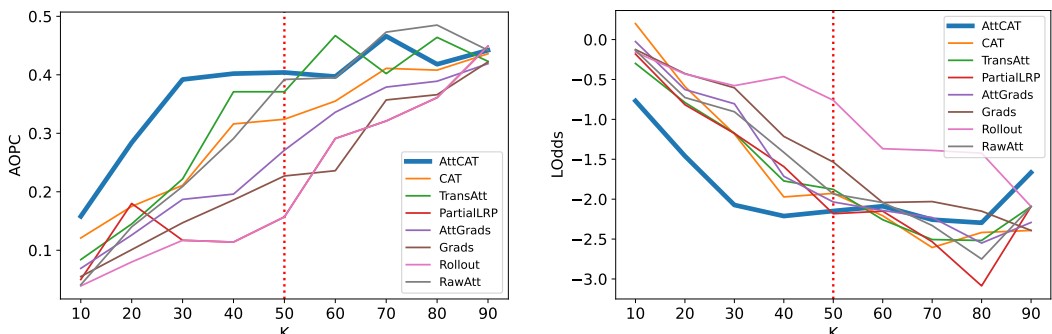

Figure 3: AOPC and LOdds scores of different methods in explaining BERT against the corruption rate $k$ on Amazon data set. Higher AOPC and lower LOdds scores are better. The x-axis demonstrates removing or masking the $k\%$ of the tokens in order of decreasing saliency.

| Method | SST2 | | QQP | | MNLI | | Amazon | | Yelp | | IMDB | |
|---|---|---|---|---|---|---|---|---|---|---|---|---|
| | AOPC↑ | LOdds↓ | AOPC | LOdds | AOPC | LOdds | AOPC | LOdds | AOPC | LOdds | AOPC | LOdds |
| RawAtt | 0.331 | -0.885 | 0.143 | 0.149 | 0.138 | 0.235 | 0.384 | -1.729 | 0.394 | -2.017 | 0.298 | -1.245 |
| Rollout | 0.286 | -0.641 | 0.139 | 0.262 | 0.151 | 0.321 | 0.324 | -1.303 | 0.277 | -1.055 | 0.331 | -1.323 |
| Grads | 0.335 | -0.252 | 0.141 | 0.184 | 0.156 | 0.139 | 0.316 | -1.820 | 0.414 | -1.994 | 0.304 | -1.227 |
| AttGrads | 0.351 | -0.603 | 0.143 | 0.113 | 0.159 | 0.114 | 0.346 | -1.941 | 0.439 | -2.054 | 0.310 | -1.267 |
| PartialLRP | 0.341 | -0.922 | 0.142 | 0.137 | 0.138 | 0.231 | 0.418 | -2.019 | 0.424 | -2.199 | 0.312 | -1.321 |
| TransAtt | 0.354 | -1.038 | **0.145** | 0.114 | 0.130 | 0.214 | 0.415 | -1.889 | 0.434 | -2.508 | 0.421 | -2.137 |
| CAT | 0.352 | -1.115 | 0.134 | 0.121 | 0.157 | 0.121 | 0.409 | -2.157 | 0.421 | -2.587 | 0.406 | -3.052 |
| AttCAT | **0.371** | **-1.319** | 0.139 | **0.073** | **0.164** | **0.008** | **0.457** | **-2.332** | **0.473** | **-3.169** | **0.528** | **-3.671** |

Table 1: AOPC and LOdds scores of different methods in explaining BERT on different data sets. Higher AOPC and lower LOdds scores are better. Best results are in bold face.

**Precision@K:** Inspired by the original Precision@K used in recommender system [45], we design a novel Precision@K to evaluate the explanation performance on SQuAD data sets. For each test example, we count the number of tokens in the answer that appear in the $K$ top-scored tokens as Precision@K. Therefore, higher Precision@K scores are better.

# 6 Results and Discussions

## 6.1 Quantitative Evaluations

The quantitative evaluations in this Section demonstrate our AttCAT method outperforms the baseline methods on the vast majority of different data sets and tasks. Table 1 depicts the results of various explanation methods and data sets. We report the average AOPC and LOdds scores over $k$ values. Due to computation costs, we experiment on a subset with 2,000 randomly selected samples for the Amazon, Yelp, and IMDB data sets. Entire test sets are used for other data sets. AttCAT achieves the highest AOPC and lowest LOdds scores in most settings, demonstrating that the most impactful tokens for model prediction have been deleted or replaced. Among all the compared methods, the attention-based methods (i.e., RawAtt and Rollout) perform worst since attention weights alone without considering the magnitudes of feature values are not adequate to analyze the inner working mechanism of Transformers. Remarkably, AttCAT also outperforms TransAtt, a recent work representing a strong baseline method. The performance of CAT, shown here as an ablation study, drops markedly, corroborating the effectiveness of using self-attention weights in AttCAT.

We also report the AOPC and LOdds scores of different methods in explaining BERT by deleting or masking bottom $k\%$ words on different data sets in Appendix Table 5. Our AttCAT achieves the lowest AOPC and highest LOdds, demonstrating that AttCAT efficiently captures the most impactful tokens for model predictions.

Figure 3 illustrates how the evaluation metrics, namely AOPC and LOdds, change over the varying corruption rates (via removing or masking the $k\%$ top-scored words). Our AttCAT method achieves the highest AOPC and the lowest LOdds scores within a corruption rate $k$ of 50% or less, further demonstrating that AttCAT has detected the most impactful words for model predictions.

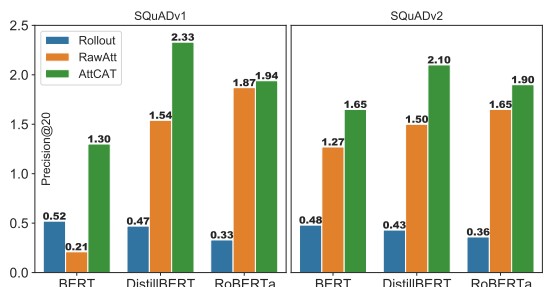 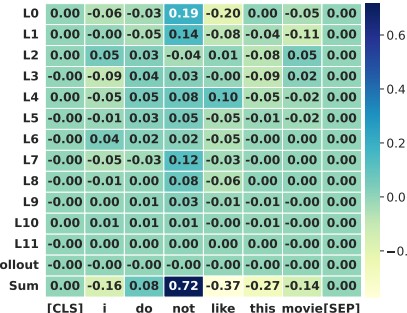

Figure 4: Precision@20 scores of the selected explanation methods for different Transformer models on SQuAD data sets. Higher scores are better. The max scores of SQuADv1 and SQuADv2 are 3.72 and 3.84, respectively.

Figure 5: Heatmap of the normalized impact scores from different BERT layers. Rollout and Sum denote the rollout and summation operations (ours), respectively. Best viewed in color.

| Method | STT2 | QQP | MNLI | Amazon | Yelp | IMDB |
|---|---|---|---|---|---|---|
| Grads | 0.150 | 0.236 | 0.169 | 0.146 | 0.174 | 0.098 |
| AttGrads | 0.116 | **0.198** | 0.156 | 0.148 | 0.132 | 0.064 |
| PartialLRP | 0.955 | 0.949 | 0.935 | 0.965 | 0.952 | 0.858 |
| TransAtt | 0.336 | 0.222 | 0.339 | 0.152 | 0.121 | 0.043 |
| CAT | 0.101 | 0.373 | 0.339 | 0.095 | 0.107 | 0.056 |
| AttCAT | **0.018** | 0.349 | **0.017** | **0.015** | **0.008** | **0.023** |

Table 2: Kendal-$\tau$ correlation of different explanation methods in explaining BERT on varying data sets. Lower scores are better. Only class-specific methods are selected. Best results are in bold face.

Table 2 shows the Kendal-$\tau$ based confidence score of the different explanation techniques for BERT tested using various data sets. We do not report the confidence scores of the attention-based methods since they are class agnostic. AttCAT achieves the best performance on most data sets; different classes observe distinctively sorted tokens, leading to much lower Kendal correlations. In other words, our AttCAT is highly confident in recognizing the most impactful tokens for predicting the class of interest.

We show the Precision@K scores for the SQuAD data sets in Figure 4. Here $k$ is set to 20. Our results clearly demonstrate that AttCAT is superior to other methods and generalizes well to various BERT architectures on SQuAD data sets. The higher score means that AttCAT can capture more impactful answer tokens in the TOP-20 sorted tokens, proving its capability to generate more faithful explanations. The results of varying $k$ values are shown in Appendix Figure 8, 9, 10, 11.

## 6.2 Qualitative Visualizations

Lastly, we show a heatmap of the normalized impact scores generated by AttCAT in Figure 5. The first 12 rows (L0-L11) show the impact scores of each token from different BERT layers. The darker shaded token represents a higher score, as shown in the legend. The signs of scores indicate their directionalities. This heatmap also justifies the effectiveness of the summation operation we used in Eq. 12. As shown in the figure, the impact scores become uniform and less impactful as the layer goes deeper, which is consistent with the observation from [13] where the authors argue that the embeddings are more contextualized and tend to carry similar information in the deeper layers. Thus, the rollout operation used in [13, 15] will attenuate the impact scores at shallower layers (i.e., L0-L9) since they are multiplied by scores at the deeper layers (i.e., L10-L11). As shown in the row of 'Rollout' in the figure, the rollout operation only gives minimal impact scores of the tokens, indicating essentially no information has been captured for the explanation. While the summation operation (ours), shown as the row of 'Sum', generates a faithful explanation incorporating the impact scores from each layer. In term of Impact Score, the token 'not' with the highest positive impact

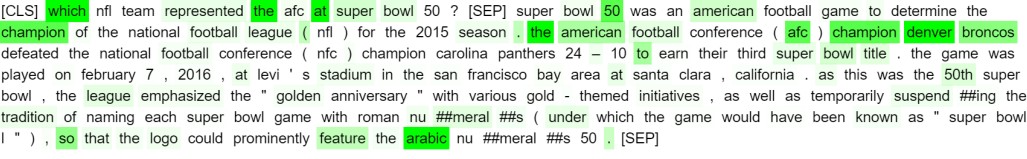

(a) A visualization of the impact scores generated by AttCAT on a showcase example in SQuAD.

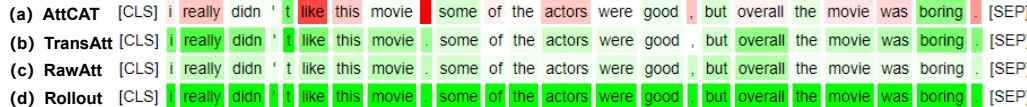

(b) Visualizations of the impact scores generated by the selected methods on a showcase example in SST2.

Figure 6: Visualization examples. The green shade indicates an important positive impact whereas the read shade means otherwise. Darker colors represent higher impact scores. Best viewed in color. More examples are shown in Appendix.

score (0.72) contributes mostly to the negative sentiment of this sentence, whereas the token 'like' with the highest negative impact score (-0.37) contributes inversely.

The ground truth answer of the question answering example shown in Figure 6a is "denver brconcos". AttCAT successfully captures these two tokens with the darkest green shades, corresponding to highest impact scores. The example from SST2 shown in Figure 6b has a negative sentiment. Both AttCAT and TransAtt capture the most impactful tokens, such as 'boring', 'didn', and 't', which contribute mostly to the negative sentiment prediction. Besides the tokens explaining the negative sentiment, our AttCAT method also identified some other tokens that contribute inversely to the negative sentiment, e.g., 'like' and 'really' (shown in dark shade of red), whereas TransAtt is not capable of differentiating positive and negative contributions. RawAtt gives more attention on some irrelevant tokens, i.e., 'overall', 'but', and the punctuations. Rollout only generates some uniformly distributed important scores for the tokens.

## 7   Conclusion

This work addresses the major issues in generating faithful and confident explanations for Transformers via a novel attentive class activation tokens approach. AttCAT leverages the features, their gradients, and corresponded attention weights to define the so-called impact scores, which quantify the impact of inputs on the model's outputs. The impact score can give both magnitude and directionality of the input tokens' impact. We conduct extensive experiments on different Transformer models and data sets and demonstrate that our AttCAT achieves the best performance among strong baseline methods using quantitative metrics and qualitative visualizations.

Even though our current AttCAT approach is mainly designed for BERT architectures on NLP tasks, it can be naturally extended to Vision Transformer architectures on computer vision tasks as the future work. Since there are various versions of Transformer architectures, e.g., ViT [3] and Swin Transformer [4], which are much different from Transformers used on NLP tasks, it opens up new avenues to extend our AttCAT to explain these models prediction.

## Acknowledgments

This work is supported by the National Science Foundation under grant IIS-2211897.

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
