# A    Transformer Models

The BERT model was initially introduced in [2]. It is a bidirectional transformer pre-trained using a combination of masked language modeling and next sentence prediction tasks on a large corpus comprising the Toronto Book Corpus and Wikipedia. We use the pre-trained BERT$_{base}$ model consisting of 12 layers and 12 multi-head self-attention modules. The hidden state size is set as 756. There are 110 million parameters in total. The DistilBERT model was proposed in [36], which is a lightweight and efficient Transformer model trained by distilling the BERT$_{base}$ model. It consists of only 6 layers and 12 multi-head self-attention modules with a hidden state size of 756, resulting in 40% less parameters than the BERT$_{base}$ model. DistilBERT runs 60% faster while preserving over 95% of BERT's performance as measured on the GLUE language understanding benchmark. Liu et al. [37] proposed the RoBERTa model, which is built on BERT with modified key hyper-parameters. It is pre-trained using the masked language modeling task with much larger mini-batches and learning rates. RoBERTa has a similar architecture as BERT but uses a byte-level BPE as a tokenizer. We report the performance of the pre-trained models from Hugging Face on various data sets and tasks in Tables 3 and 4. These results demonstrate that all these pre-trained models that we used for generating their explanations achieve good performance on the corresponding tasks.

| Datasets | SST2 | QQP | MNLI | Amazon | Yelp | IMDB |
|---|---|---|---|---|---|---|
| BERT$_{base}$ | 92.43 | 92.40 | 84.00 | 94.65 | 96.30 | 91.90 |

Table 3: Accuracies of BERT$_{base}$ model on test sets of the selected tasks.

| Dataset | SQuADv1 | | SQuADv2 | |
|---|---|---|---|---|
| Metric | EM | F1 | EM | F1 |
| BERT$_{base}$ | 80.90 | 88.20 | 76.54 | 83.00 |
| DistilBERT | 79.12 | 86.90 | 75.14 | 81.50 |
| RoBERTa | 83.00 | 90.40 | 79.87 | 82.91 |

Table 4: Evaluation results of the selected Transformer models on the question answering task. Exact match (EM) measures the percentage of predictions that match any one of the ground truth answers exactly. Macro-averaged F1 score (F1) measures the average overlap between the predicted and ground truth answers. F1 is computed by treating the predicted and ground truth answers as bags of tokens.

# B    Tasks and Datasets

**Sentiment Analysis:** The Stanford Sentiment Treebank (SST2) [38] and IMDB [40] consist of sentences extracted from movie reviews with human annotations of the sentiments. They are designed to predict the sentiment score for a given sentence in a binary scale. The Amazon and Yelp data sets consist of reviews from Amazon and Yelp, respectively. The polarity data sets, which consider stars 1&2 as negative and stars 3&4 as positive, are constructed by [39] for binary sentiment classification.

**Natural Language Inference:** The Multi-Genre Natural Language Inference Corpus (MNLI) [41] is a crowd-sourced collection of sentence pairs with textual entailment annotations. Given a premise sentence and a hypothesis sentence, the task is to predict whether the premise entails the hypothesis (entailment), contradicts the hypothesis (contradiction), or neither (neutral).

**Paraphrase Detection:** The Quora Question Pairs2 (QQP) [42] is a collection of question pairs from the community question-answering website Quora. The task is to determine whether a pair of questions are semantically equivalent.

**Question Answering:** The Stanford Question Answering (SQuADv1 [43] and SQuADv2 [44]) are reading comprehension data sets consisting of questions posed by crowd workers on a set of Wikipedia articles. The answer to every question is a segment of text, or span, from the corresponding reading passage, or the question is unanswerable.

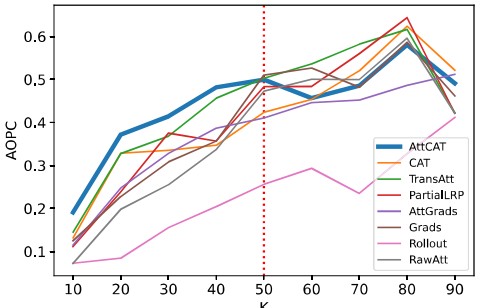 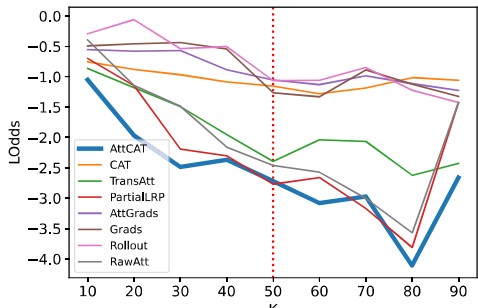

Figure 7: AOPC and LOdds scores of different methods in explaining BERT across the varying corruption rates $k$ on Yelp data set. The x-axis demonstrates removing or masking the $k\%$ of the tokens in an order of decreasing saliency.

| Method | SST2 | | QQP | | MNLI | | Amazon | | Yelp | | IMDB | |
|---|---|---|---|---|---|---|---|---|---|---|---|---|
| | AOPC↑ | LOdds↓ | AOPC | LOdds | AOPC | LOdds | AOPC | LOdds | AOPC | LOdds | AOPC | LOdds |
| RawAtt | 0.164 | -0.516 | 0.139 | 0.251 | 0.218 | 0.178 | 0.118 | -0.262 | 0.142 | -0.697 | 0.148 | -0.457 |
| Rollout | 0.212 | -0.675 | 0.132 | 0.278 | 0.206 | 0.105 | 0.157 | -0.358 | 0.152 | -0.613 | 0.110 | -0.465 |
| Grads | 0.214 | -0.717 | 0.122 | 0.214 | 0.208 | 0.195 | 0.119 | -0.214 | 0.126 | -0.624 | 0.115 | -0.501 |
| AttGrads | 0.210 | -0.706 | 0.125 | 0.199 | 0.210 | 0.186 | 0.123 | -0.206 | 0.118 | -0.538 | 0.112 | -0.524 |
| PartialLRP | 0.159 | -0.475 | 0.519 | 0.281 | 0.218 | 0.190 | 0.151 | -0.290 | 0.146 | -0.711 | 0.142 | -0.515 |
| TransAtt | 0.139 | -0.324 | **0.045** | 0.283 | 0.211 | 0.239 | 0.115 | -0.192 | 0.111 | -0.359 | 0.134 | -0.308 |
| CAT | 0.207 | -0.458 | 0.134 | 0.276 | 0.216 | 0.180 | 0.117 | -0.201 | 0.124 | -0.526 | 0.142 | -0.323 |
| AttCAT | **0.121** | **-0.320** | 0.112 | **0.326** | **0.183** | **0.245** | **0.108** | **-0.039** | **0.098** | **0.025** | **0.088** | **-0.154** |

Table 5: AOPC and LOdds scores of various methods in explaining BERT model prediction on different data sets. Lower AOPC and higher LOdds scores are better. Best results are in bold face.

## C    Additional Results

We report the AOPC and LOdds scores of different methods in explaining BERT model prediction by deleting or masking bottom $k\%$ words on different data sets in Table 5. Our AttCAT achieves the lowest AOPC and highest LOdds, further demonstrating that AttCAT efficiently captures the most impactful tokens for model predictions.

Figure 7 illustrates how the evaluation metrics, namely AOPC and LOdds, change over the varying corruption rates (via removing or masking the $k\%$ top-scored words) on Yelp data set. Our AttCAT achieves the highest AOPC and the lowest LOdds scores with the corruption rate $k$ of 50% or less, further demonstrating that AttCAT's capability of detecting the most important words for model predictions.

We show the Precision@K (i.e., $K = 10, 30, 40, 50$) scores of the selected explanation methods for various Transformer models on SQuAD data sets in Figure 8, 9, 10, 11. The max scores of SQuADv1 and SQuADv2 are 3.72 and 3.84, respectively. These results further demonstrate that our AttCAT outperforms other baselines on various models. The higher scores mean that AttCAT can generate more faithful explanations with different $K$ values. Especially, AttCAT outperforms others in a largest margin in terms of Precision@10 score, demonstrating that it captures the most impactful answer tokens in the TOP-10 sorted tokens shown in Figure 8.

All the examples from SST2 shown in Figure 12 present a positive sentiment. Our method AttCAT captures the most impactful tokens, such as 'fresh', 'like', 'thanks', and 'compelling' (shown in dark shade of green), which contribute mostly to the positive sentiment prediction. Besides these positive tokens, the AttCAT method also identifies other tokens that contribute inversely to the positive sentiment, e.g., 'neither' and 'nor' (shown in dark shade of red). However, TransAtt is not capable of differentiating positive and negative contributions. Both RawAtt and Rollout methods seemingly generate uniformly distributed importance scores for the tokens.

The example from the Yelp Polarity data set shown in Figure 13 has a positive sentiment. Both our AttCAT and TransAtt methods are able to capture the most important tokens for the positive sentiment prediction, such as 'better', 'best', and 'worth'. However, the TransAtt method also generates higher scores for some irrelevant tokens, i.e., 'am', 'always', and 'selection'. RawAtt and

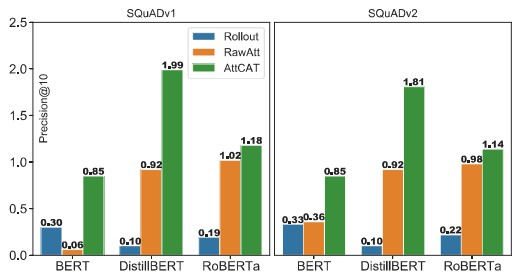

Figure 8: Precision@10 scores.

Figure 9: Precision@30 scores.

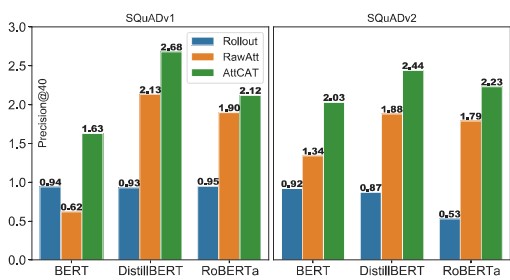

Figure 10: Precision@40 scores.

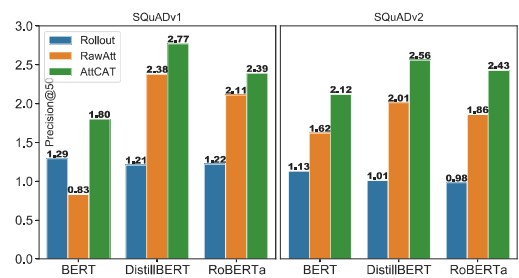

Figure 11: Precision@50 scores.

Rollout only generate the uniformly distributed importance scores for the tokens, demonstrating unfaithful explanations.

The ground truth answer of the question answering example shown in Figure 14 is "nevada". AttCAT successfully captures this token with the darkest green shade, corresponding to the highest impact score. Nevertheless, all other baselines fail to give faithful explanations for this question answering task directly.

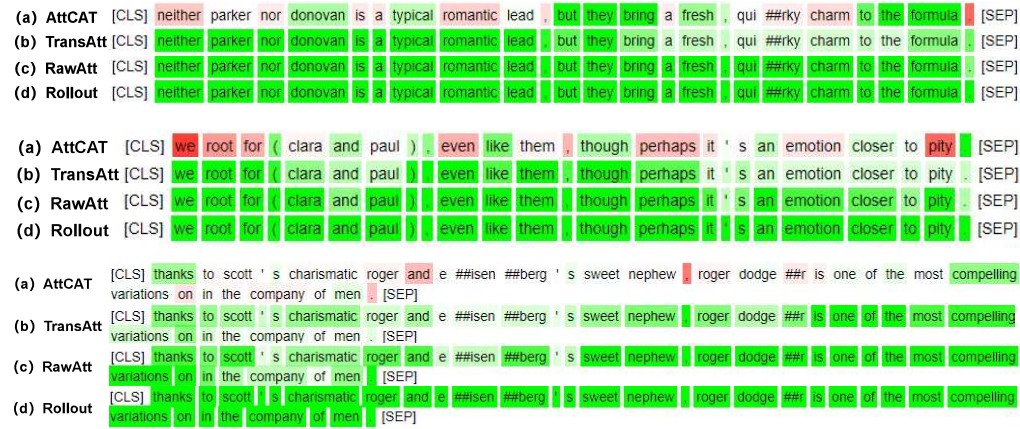

Figure 12: Visualization of impact scores generated by the selected methods in three showcase examples of SST2 on sentiment analysis task.

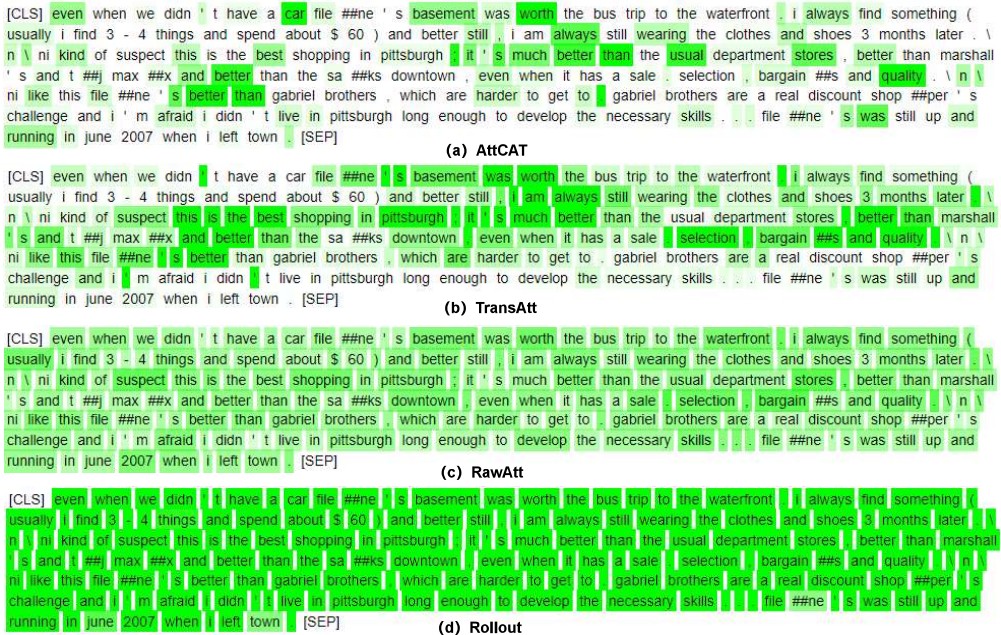

Figure 13: Visualizations of the impact scores generated by the selected methods of a showcase example in Yelp Polarity on sentiment analysis task.

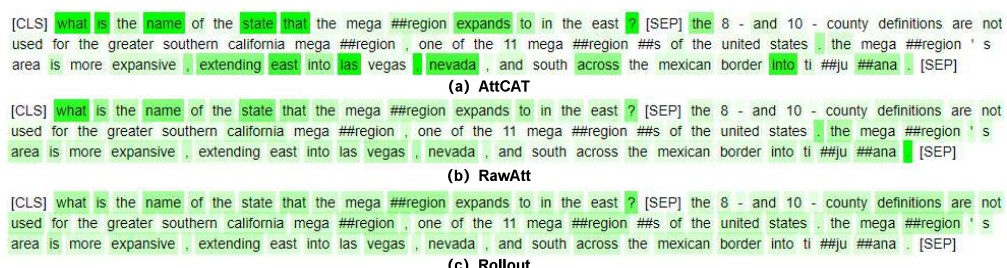

Figure 14: Visualizations of the impact scores generated by the selected methods of a showcase example in SQuAD on Q&A task.