# OpenReview forum: "AttCAT: Explaining Transformers via Attentive Class Activation Tokens"
_NeurIPS.cc/2022/Conference — NeurIPS 2022 Accept_

### Official Review · Reviewer_HxzL · 2022-06-30

**Rating:** 5
**Confidence:** 3
**Soundness:** 3 good
**Presentation:** 1 poor
**Contribution:** 2 fair

**Summary:**

This paper proposed two methods to understand how transformers work: CAT and AttCAT. They are motivated to take care of the magnitudes of the features, the gradients, and the skip connection to examine (1) which tokens mostly influence the model's output and (2) whether a token pose a positive or negative contribution to the output. Both CAT and AttCAT are developed based on GradCAM. It also studied a lot of previous approaches and mainly compared weight-based methods, gradient-based methods, and methods based on layer-wise relevance propagation. Results on diverse common benchmarks show the strength of the proposed method, including SST2, QQP, MNLI, Amazon, Yelp, and IMDB. Some qualitative results are included to help understand.

**Questions:**

The reviewer is mainly concerned with the missing of potential ablation studies and will adjust scores if it gets addressed properly.

**Limitations:**

The reviewer currently did not see sufficient ablation studies to support their claims. Also, the method currently did not provide evidence to support understanding of how transformers work in vision tasks.

**Strengths And Weaknesses:**

The reviewer believes the studied direction is pretty much important for related communities where transformers gradually become the most common techniques we used in various tasks. The reviewer also agrees with the authors that we should also consider the magnitude of the features and the skip connection which play big roles in transformers.

They carefully studied a lot of literature and mainly compared with three categories including: (1) attention-weights-related works, like RawAtt and Rollout; (2) gradient-related works, like Grads and AttGrads; (3) layer-wise relevance propagation methods, like PartialLRP and TransAtt. Figure2 clearly tells the difference between the proposed method and all the previous approaches.

Extensive quantitative experiments are conducted over various datasets including SST2, QQP, MNLI, Amazon, Yelp, IMDB, and SQuAD v1 and v2. Results show the proposed method generally achieves better performance compared with the previous approaches. The reviewer especially likes the red notation in the visualization of the proposed method which indicates the negative effect of the input texts.


The paper currently fused the introduction and related works into one section, which makes the introduction section pretty lengthy and a bit hard to read. Although the authors described the differences between the proposed method and previous methods in many places (intro, method, exps), the reviewer did not clearly understand the advantages of the proposed method and the key point leading to the good performance.

Also, the reviewer did not see related ablation studies related to two major claims that we should exploit (1) the magnitudes of the features and (2) the skip connections. There are huge missings, where the authors can not consider the skip connections when aggregating the information to see if considering skip connections really help the understanding.

Besides, the abstraction shows the reviewer the proposed method address various different tasks while it mainly focused on NLP tasks and currently have no discussion to tasks in other domains.

---

> ### Author Response · Authors · 2022-08-01
> **Response to Reviewer HxzL**
>
> **"The paper currently fused the introduction and related works into one section…  the reviewer did not clearly understand the advantages of the proposed method and the key point leading to the good performance. "**
>
> Thank you for your comment. The reason why we fuse these two sections is that we want to seamlessly survey the literature on explaining Transformer and describe the current research gaps, collectively illustrated in Figure 1. We also highlight the formulation differences between the baseline methods and our proposed method in Figure 2, as pointed out by the reviewer. In Figure 2, we clearly observe that our methods AttCAT and CAT (ablation version) leverage the features h and their gradients, demonstrating the point that our method exploits the magnitudes of the features directly. Additionally, since the features h are directly aggregated from two parts, as shown in Figure 1 and Eq. 2, our method utilizes the skip connection directly to generate the explanations.
>
> **Ablation studies on (1) the magnitudes of the features and (2) the skip connections**
>
> Thanks for the valuable suggestion. In the original manuscript, we have conducted one ablation study, which compares our AttCAT and CAT (ablation version) to see whether attention weights help generate better explanations. In addition, we may consider other baseline methods as ablation studies on the magnitude of features (the first ablation studies suggested by the reviewer). Compared to our methods, RawAtt and Rollout only exploit the attention weights without features and their gradients. Similarly, Grads and AttGrads only exploit the attention weights and their gradients without considering the features and their gradients. PartialLRP and TransAtt only exploit the attention weights gradients and layer-wise relevance propagation. For another ablation study suggested by the reviewer on studying the effect of the skip connections, it seems unnecessary to do the ablation study on the skip connections separately since we do not want to explain a Transformer with a major information loss. It is because the skip connection component is a core of feature aggregation, as shown in Eq. 2. Moreover, as mentioned in Lu et al. [1] that a significant portion of information flow in BERT goes through the skip connection instead of the attention heads (i.e., three times more often than attention on average).
>
> **Tasks in other domains**
>
> We thank the reviewer for pointing out the application of our method in computer vision tasks. In this paper, we mainly develop our method to explain Transformers (i.e., BERT base model and variants such as DistillBERT and RoBERTa) on various NLP tasks, such as sentiment analysis (SST2, IMDB, Yelp, and Amazon), natural language inference (MNLI), paraphrase detection (QQP), and question answering (SQuADv1 and SQuADv2). We also present some details of these tasks in Section B of Appendix. Since there are different visions of Transformer architectures, such as ViT [2] and Swin Transformer [3], which are much different from Transformers used on NLP tasks, it is valuable to explain these architectures and extend our method as our future works.
>
> **References:**
>
> [1] Kaiji Lu, Zifan Wang, Piotr Mardziel, and Anupam Datta. Influence patterns for explaining information flow in bert. Advances in Neural Information Processing Systems, 34, 2021
>
> [2] Dosovitskiy, Alexey, Lucas Beyer, Alexander Kolesnikov, Dirk Weissenborn, Xiaohua Zhai, Thomas Unterthiner, Mostafa Dehghani et al. "An image is worth 16x16 words: Transformers for image recognition at scale." arXiv preprint arXiv:2010.11929 (2020).
>
> [3] Liu, Ze, Yutong Lin, Yue Cao, Han Hu, Yixuan Wei, Zheng Zhang, Stephen Lin, and Baining Guo. "Swin transformer: Hierarchical vision transformer using shifted windows." In Proceedings of the IEEE/CVF International Conference on Computer Vision, pp. 10012-10022. 2021.

---

> > ### Comment · Reviewer_HxzL · 2022-08-08
> > **Reviewer HxzL response**
> >
> > I appreciate the efforts made by the authors for the detailed response.
> >
> > Regarding the skip connections, I feel it is more about self-contained. It is important to check if your conclusion has the same trend as the previous works showed.
> >
> > Regarding the task in other domains, if you mainly tested the proposed method in NLP tasks, it is probably good to articulate the words as you showed to me here.

---

> > > ### Author Response · Authors · 2022-08-09
> > > **Response to Reviewer HxzL about skip connections:**
> > >
> > > Thank you for taking time to read our response! We appreciate your suggestion on articulating the possible extension of our AttCAT to other domains, and we will carefully incorporate our response in the final version of this manuscript.
> > >
> > > Here we would like to provide more information on the point you raised regarding ablation studies of skip connections.
> > >
> > > Several existing works [1][2][3] including ours (as shown in Figure 1 and Eq. 2) demonstrate that the skip connections are the crucial component of Transformer. Generating explanations without the skip connections may not duly demonstrate the inner working of Transformers due to the significant information loss. It is worth mentioning that Lu et al. NeurIPS 2021 [1] have done the ablation study on skip connections (Table 1 in their paper) and concluded that the ablation only produces random ablated accuracies on various NLP tasks, including Subject-Verb Agreement (SVA), Reflexive Anaphora (RA), and Sentiment Analysis (SA). This ablation study further substantiates the pivotal role of skip connections for Transformers to function correctly.
> > >
> > > We elaborate on the details of related works below:
> > >
> > > Lu et al. NeurIPS 2021 [1] demonstrate that a significant portion of information flow in BERT goes through the skip connections instead of the attention heads. Furthermore, the authors discuss that the important information is simply “copied” to the next layer through the skip connections. Thus the skip connections are traversed much more often than attention heads. This is consistent with our analysis of disentangling information flows in Transformer in Section 4.1. Importantly, their ablation study on skip connections corroborates that skip connections relay important information directly and cannot be replaced or dropped out.
> > >
> > > Brunner et al. ICLR 2020 [2] first point out that there is a strong influence passing through skip connections, which retains the identity information of the input tokens. The identity information refers to the ability of a model to learn stable representations, which is a desirable property affecting the replicability and interpretability of the Transformer’s predictions. Thus, the ablation study on the skip connections may cause identity information loss, leading to poor interpretability.
> > >
> > > Similarly, Dong et al. ICML 2021 [3] show the importance of skip connections by decomposing and analyzing forward-pass computations in self-attention modules.
> > >
> > > References:
> > >
> > > [1] Kaiji Lu, Zifan Wang, Piotr Mardziel, and Anupam Datta. Influence patterns for explaining information flow in bert. Advances in Neural Information Processing Systems, 34, 2021
> > >
> > > [2] Brunner, Gino, Yang Liu, Damian Pascual, Oliver Richter, Massimiliano Ciaramita, and Roger Wattenhofer. "On identifiability in transformers." ICLR (2020).
> > >
> > > [3] Dong, Yihe, Jean-Baptiste Cordonnier, and Andreas Loukas. "Attention is not all you need: Pure attention loses rank doubly exponentially with depth." In International Conference on Machine Learning, pp. 2793-2803. PMLR, 2021.

---

### Official Review · Reviewer_1Khh · 2022-07-10

**Rating:** 8
**Confidence:** 3
**Soundness:** 4 excellent
**Presentation:** 4 excellent
**Contribution:** 3 good

**Summary:**

The paper proposes "Attentive Class Activation Tokens" (AttCAT), a post-hoc method for the explanation of Transformer models addressing NLP tasks.

Different from existing related methods which only focus specific components of the models being explained (leading to reduced faithfulness) and/or heuristics, the proposed method stresses the use of a) the features encoded by the model, b) their gradients, and c) their associated attention weights as a complete package to address the weaknesses in existing explanation methods.
This enables not only the explanation of the parts of the input (tokens) that have a high impact on the prediction made by the model, but also whether this impact positively or negatively contributed to the prediction (directionality).

**Questions:**

N.A.

**Limitations:**

N.A.

**Strengths And Weaknesses:**

= Strengths

+ The manuscript had a very good structure and organization. This led to clear content with a good flow. Overall I enjoyed reading this paper. I applaud the authors for the effort put on the presentation of this manuscript.

+ The proposed method is relatively simple. The inclusion of each of the components (the features encoded by the model, their gradients, and their associated attention weights) is well motivated.

+ On the qualitative side, to the best of my knowledge, the proposed method is novel and complementary to what is out there. On the quantitative side, it obtained state-of-the-art results w.r.t. existing methods.

+ Empirical validation of the method was conducted considering a rich set of well known components including several transformer architectures, several NLP-related datasets, and several model explanation baselines from the literature.

+ Observations made via the proposed method align with observation from previous efforts, e.g. [11] in Sec. 6.2. This already hints at the added value of the proposed method.

---

> ### Author Response · Authors · 2022-08-01
> **Response to Reviewer 1Khh**
>
> We thank Reviewer 1Khh for summarizing the strengths and novelty of our work from both the qualitative and quantitative sides, as well as the structure of our presentation and organization. Particularly, we appreciate your insight on the alignment of our work and previous ones and highlight our original contributions. As mentioned in your comment, we quantitatively show our sum operation is better than the rollout operation in Figure 4 of Section 6.2.

---

### Official Review · Reviewer_ra8V · 2022-07-11

**Rating:** 6
**Confidence:** 4
**Soundness:** 3 good
**Presentation:** 3 good
**Contribution:** 3 good

**Summary:**

The paper proposes a new method, termed AttCAT, for evaluating the importance of each input token in Transformers. The proposed method considers two perspectives for token importance, including the attention perspective and the gradient perspective. The authors introduce Class Activation Tokens, which is inspired by GradCAM [26]. And the Class Activation Tokens is combined with the Transformer attention to output AttCAT. The experimental results show that the resulting metric for token importance is superior to some other methods.

**Questions:**

- Instead of simply averaging the AttCAT over multiple heads and multiple layers, are there better methods for doing the aggregation of AttCAT here?
- Could the author provide the statistics of the datasets, e.g., number of classes and dataset size?

**Limitations:**

The authors mention that they would extend the AttCAT method to explain generative and vision Transformer architectures as future works. But more discussions on limitations of this work would be useful.

**Strengths And Weaknesses:**

## Strengths
- The method is well motivated and clearly stated. It makes sense to combine the attention scores and the gradient weights to obtain a new metric for token importance in Transformers.
- The experimental results clearly demonstrate the effectiveness of the proposed metric.

## Weaknesses
- The experiments could be extended. It would be helpful to see how the evaluation metrics, namely AOPC and LOdds, change against the corruption rate $k$ (removing the $k$% top-scored words). Also, it is interesting to see how the two metrics change when we remove the  $k$% lowest scored words. This is different from removing the $k$% top-scored words because the inputs to the model are different in the two cases. When removing the $k$% lowest scored words, we are caring the order of the less informative tokens, while in removing the $k$% lowest scored words, we are caring the order of the most informative tokens.

---

> ### Author Response · Authors · 2022-08-01
> **Response to Reviewer ra8V**
>
> We appreciate the reviewer’s comment on examining performance change against the corruption rate. We have performed a similar analysis in the original manuscript. As we stated in Section 5.3 (lines 245 and 246, "To avoid choosing an arbitrary **k** , we remove 0, 10, 20, · · · , 100% of the tokens in order of decreasing saliency, thus arriving at… ”. We report the average AOPC and LOdds scores over varying values. We have added some results of the Amazon dataset in the following table. Our AttCAT achieves the highest AOPC scores with a small corruption rate  **k**  (i.e., 10, 20, 30), further demonstrating that AttCAT has detected the most important words for model predictions.
>
> | Method     | 10    | 20    | 30    | 40    | 50    | 60    | 70    | 80    | 90    |
> |------------|-----------|-----------|-----------|-------|-------|-------|-------|-------|-------|
> | RawAtt     | 0.041     | 0.140     | 0.209     | 0.291 | 0.392 | 0.395 | 0.473 | 0.485 | 0.442 |
> | Rollout    | 0.039     | 0.080     | 0.117     | 0.114 | 0.157 | 0.291 | 0.321 | 0.361 | 0.449 |
> | Grads      | 0.055     | 0.101     | 0.147     | 0.186 | 0.227 | 0.236 | 0.357 | 0.366 | 0.422 |
> | AttGrads   | 0.069     | 0.126     | 0.187     | 0.196 | 0.271 | 0.336 | 0.379 | 0.389 | 0.419 |
> | PartialLRP | 0.050     | 0.180     | 0.117     | 0.114 | 0.157 | 0.291 | 0.321 | 0.361 | 0.449 |
> | TransAtt   | 0.084     | 0.145     | 0.222     | 0.371 | 0.371 | 0.467 | 0.402 | 0.464 | 0.423 |
> | CAT        | 0.121     | 0.175     | 0.211     | 0.316 | 0.324 | 0.355 | 0.411 | 0.408 | 0.436 |
> | AttCAT     | **0.158** | **0.284** | **0.392** | 0.402 | 0.404 | 0.397 | 0.466 | 0.418 | 0.442 |
>
> We also agree with the reviewer that it is also useful to see how the two metrics change when we remove the  **k** % lowest scored words. We have added the results in the table below. Since we are removing the  **k** % lowest words, which are the less informative tokens, lower AOPC and higher LOdds scores are better. Our AttCAT achieves the best performance on Amazon and Yelp datasets shown in the table.
>
> | Method     | Amazon | Amazon | Yelp  | Yelp   |
> |------------|--------|--------|-------|--------|
> |            | AOPC   | LOdds  | AOPC  | LOdds  |
> | RawAtt     | 0.118  | -0.262 | 0.142 | -0.697 |
> | Rollout    | 0.157  | -0.358 | 0.152 | -0.613 |
> | Grads      | 0.119  | -0.214 | 0.126 | -0.624 |
> | AttGrads   | 0.123  | -0.206 | 0.118 | -0.538 |
> | PartialLRP | 0.151  | -0.290 | 0.146 | -0.711 |
> | TransAtt   | 0.115  | -0.192 | 0.111 | -0.359 |
> | CAT        | 0.117  | -0.201 | 0.124 | -0.526 |
> | AttCAT     | **0.108**  | **-0.039** | **0.098** | **0.025**  |
>
> We would add others results in our final version.
>
> **Q1 “Instead of simply averaging the AttCAT over multiple heads and multiple layers, are there better methods for doing the aggregation of AttCAT here?”**
>
> Thank you for giving this thoughtful suggestion. There are an array of works discussing how we can deal with the multiple heads and multiple layers. They are either pruning [1] (or ablating [2]) several heads or probing a single head or layer [3]. Since we are explaining the performance of the pre-trained Transformers, it might not be a good idea to prune or ablate attention heads or layers due to the information loss. In addition, probing of a single attention head or layer may be insufficient to explain the inner working mechanism of Transformers. Existing works, e.g., [4][5], typically average over the multiple heads and/or rollout over the multiple layers. Yet the rollout also triggers information loss issues in some cases, as we demonstrated in Figure 4, which motivated us to utilize a summation operation over the multiple layers. Although the results are promising, we will keep investigating better methods for aggregating over multiple heads in future work.
>
> **Q2 Datasets Statistics**
>
> | Datasets | #Test Samples | # Classes |
> |----------|---------------|-----------|
> | SST2     | 1,821         | 2         |
> | QQP      | 2,000         | 2         |
> | MNLI     | 9,815         | 2         |
> | Amazon   | 2,000         | 2         |
> | Yelp     | 2,000         | 2         |
> | IMDB     | 2,000         | 2         |
>
> References:
>
> [1] Voita, Elena, David Talbot, Fedor Moiseev, Rico Sennrich, and Ivan Titov. "Analyzing multi-head self-attention: Specialized heads do the heavy lifting, the rest can be pruned." arXiv preprint arXiv:1905.09418 (2019).
>
> [2] Michel, Paul, Omer Levy, and Graham Neubig. "Are sixteen heads really better than one?." Advances in neural information processing systems 32 (2019).
>
> [3] Clark, Kevin, Urvashi Khandelwal, Omer Levy, and Christopher D. Manning. "What does bert look at? an analysis of bert's attention." arXiv preprint arXiv:1906.04341 (2019).
>
> [4] Samira Abnar and Willem Zuidema. Quantifying attention flow in transformers. arXiv preprint arXiv:2005.00928, 2020.
>
> [5] Hila Chefer, Shir Gur, and LiorWolf. Transformer interpretability beyond attention visualization. CVPR 2021.

---

> > ### Comment · Reviewer_ra8V · 2022-08-10
> > **Comments to rebutal**
> >
> > The authors addressed my concerns, so I would like to keep my rating.

---

### Official Review · Reviewer_pvDc · 2022-07-12

**Rating:** 4
**Confidence:** 3
**Soundness:** 2 fair
**Presentation:** 3 good
**Contribution:** 3 good

**Summary:**

The paper proposes a Transformer explanation technique via attentive class activation tokens, aka, AttCAT, leveraging encoded features, their gradients, and their attention weights to generate a faithful and confident explanation for Transformer’s output.


**Questions:**

In Table 1, when evaluating on QQP, TransAtt seems to have a better evaluation performance than the proposed method, AttCAT. Is there a reason why this happens?

Similar pattern can be observed in Table 2, AttGrads also has a better performance than the proposed method, AttCAT. Does this mean the evaluation performance heavily depends on the dataset and may not be consistent? Therefore, is the improvement of AttCAT also conditional?

In the qualitative comparison, the author does not compare AttCAT against CAT. I think this is the most important comparison, especially when the difference between AttCAT and CAT is very limited.


**Ethics Review Area:**

["I don’t know"]

**Limitations:**

The work does not talk about its limitations.

Especially refer to above, it would be important to know in which dataset, AttCAT is better than former ones and in which dataset similar to QQP AttCAT may not perform better.

**Strengths And Weaknesses:**

The work clearly defines the problem and presents a detailed work-through with formulas of the proposed method.

The work also present experiments with obvious gains to demonstrate the effectiveness of the proposed method

The proposed method seems to have limited novel differences against the former method, CAT.
The work needs more qualitative and quantitative evaluation methods to prove why AttCAT is better than former methods in helping people understand Transformers instead of just showing performance numbers.

---

> ### Author Response · Authors · 2022-08-01
> **Response to Reviewer pvDc**
>
> **“The proposed method seems to have limited novel differences against the former method, CAT.”**
>
> Thanks for bringing up this important question, but we want to reemphasize that CAT is also a new method that we proposed in this work, which can be considered as an ablation version of AttCAT without incorporating attention weights. In Figure 2 of the original manuscript, the purple box clearly shows the differences between CAT and AttCAT in the formulation.
>
> **“The work needs more qualitative and quantitative evaluation methods to prove why AttCAT is better than former methods in helping people understand Transformers instead of just showing performance numbers.”**
>
> Thanks for your comment. As we presented in Section I (lines 78-88) and Section 4 (lines 191-197), our method demonstrates a better performance than the baseline methods since we leverage not only the magnitude of the features and the skip connections but also the attention weights to interpret the inner working mechanism of Transformers. The effectiveness of our method was shown via both qualitative (i.e., Tables 1&2 and Figures 3&4) and quantitative (i.e., Figure 5) evaluations. More evaluations are presented in our Appendix, including 1) precision with varying k values and 2) visualization of impact score over various methods and tasks.
>
> **Questions 1&2 "In Table 1, when evaluating on QQP, TransAtt seems to have a better evaluation performance than the proposed method, AttCAT. Is there a reason why this happens" and "Similar pattern can be observed in Table 2, AttGrads also has a better performance than the proposed method, AttCAT. Does this mean the evaluation performance heavily depends on the dataset and may not be consistent? Therefore, is the improvement of AttCAT also conditional?"**
>
> Thank you for your comment. We understand where you are from, but we would like to emphasize that our AttCAT method outperforms the baseline methods on the vast majority of different data sets and tasks. We note that the questions in the QQP dataset are typically very short, with only a few words. In other words, it is easy to capture the most important words in the QQP task within short sentences. As a result, all the compared methods achieve a good performance. On the contrary, other datasets have longer sentences with more complex structures, which demonstrates that our method is superior to the baseline methods in more complex data sets and tasks, as shown in Tables 1 and 2.
>
> **Questions 3 "In the qualitative comparison, the author does not compare AttCAT against CAT. I think this is the most important comparison, especially when the difference between AttCAT and CAT is very limited."**
>
> As we reemphasized before, CAT, as one of this work’s contributions, is an ablation version of AttCAT without incorporating attention weights. We observe that its performance drops compared with AttCAT in the quantitative evaluations, so we only select AttCAT as our final version to qualitatively compare with other baseline methods.

---

### Author Response · Authors · 2022-08-01
**Thank you, all reviewers.**

Thank you all for the time you took to review our paper. We hope that our responses have fully addressed your concerns and remain committed to clarifying any further questions that may arise during the discussion period.

---

### Meta-Review · Area_Chair_SCAp · 2022-08-25

**Recommendation:** Accept
**Confidence:** Certain

**Metareview:**

This is an interesting paper with good contribution to the field. Most reviews are positive.

**Award:**

No

---

### Decision · Program_Chairs · 2022-09-14

Accept